# Conceptualizing Festival Attractiveness and Its Impact on Festival Hosting Destination Loyalty: A Mixed Method Approach

**Jing Li** [1], **Guangquan Dai** [1,*] , **Jinwen Tang** [1] **and Ying Chen** [2]

1    School of Economics and Commerce, South China University of Technology, Guangzhou 510006, China;
     eclijing75@mail.scut.edu.cn (J.L.); 201510106118@mail.scut.edu.cn (J.T.)
2    Marketing Department, Guangdong Communication Polytechnic, Guangzhou 510650, China;
     chenying@gdcp.cn
*    Correspondence: gqdai@scut.edu.cn; Tel.: +86-138-2619-4271

**Abstract:** A number of studies have been conducted to examine the attractiveness of tourism destinations. However, there has been little research done on festival attractiveness nor on its impact on destination loyalty. This study employed a mixed method approach to investigate the festivals in Guangzhou city, China. Firstly, through a qualitative method, the four dimensions of festival attractiveness were identified, i.e., strong festival atmosphere, harmonious interpersonal interaction, distinct cultural symbols and rich festival activities. Then, based on a cognitive–affective–conative model, a quantitative method was used to explore the mechanism through which festival attractiveness impacts sense of place and destination loyalty. Structural equation modeling showed that all dimensions of festival attractiveness have positive effects on place attachment, but not all of them have positive effects on place identity. Findings further indicate that place attachment has a positive effect on place identity, and that place attachment and place identity both have positive effects on destination loyalty. These results indicate that festival hosting destination loyalty follows the processes of both festival attractiveness cognition and destination affection evaluation.

**Keywords:** festival attractiveness; destination loyalty; sense of place; mixed method; cognitive-affective-conative model

## 1. Introduction

The recent growth in the quantity, diversity and popularity of festivals [1] has made them one of the fastest growing segments in the tourism industry [2,3], which has attracted worldwide recognition and attention. Getz pointed out that the driving force behind festival tourism lies in its attractiveness [4]. So far, researches have mainly focused on the attractiveness of different tourism destinations, such as the attractiveness of hot spring destinations [5], the attractiveness of exhibition destinations [6] and the attractiveness of wellness destinations [7], as well as their measurements and constitutive dimensions. Some researches treated festivals as one of the dimensions of destination attractiveness [8,9] but neglected festival attractiveness itself.

Attractiveness is an important perception during tourists' participation in tourism activities [10]. Tsaur et al. pointed out that festival attractiveness is a critical factor influencing tourist motivation and participation in activities [11]. Meanwhile, festival attractiveness can enhance tourists' intention to revisit [12], and it is the key to the success of festival activities. Moreover, the research on festival attractiveness can help organizers create an attractive thematic environment through rites, atmosphere, services and interactive activities, but also helps attract tourists into activities, thus enhancing the diversity of their experience. However, previous studies have not identified the dimensions of festival

attractiveness, failing to advance the theoretical research on festivals and provide reference for festival management. Therefore, it is necessary to explore the constitutive dimensions of festival attractiveness from the perspectives of both organizers and tourists.

Festivals help protect local cultural traditions, develop tourism and promote the economic, social and cultural developments of the destination [13,14]. Specifically, the success of a festival helps attract large crowds, thus strengthening the attractiveness of cities, communities or tourism destinations. In addition, they promote the economic development of the destination, providing more job opportunities [15]. They also enable tourists from different places to be exposed to and understand unique cultural heritages, as well as ethnic and local customs [16,17]. It is also believed that festivals can attract more tourists to enrich tourist sources, and that they can make tourists stay longer to achieve a balance between hot seasons and slack seasons [17]. From a global perspective, many tourism destinations use festivals as an important means to attract tourists, making festivals an important part of the tourism product to attract tourists from all over the world. Therefore, there is an inextricable relationship between festival attractiveness and hosting destination.

As an important tourist attraction, festivals can enhance people–place interactions through diversified activities, so as to influence tourist loyalty [18]. In fact, many factors can influence tourist loyalty, such as destination attachment [19], festival satisfaction [20] etc. Especially, Yuksel, Yuksel and Bilim found that a moderate level of emotional attachment can arise among tourists who are satisfied with the festival and their loyalty to the destination could thus be enhanced [19]. This kind of emotional attachment to the destination is called sense of place, which is an affective connection that results from people–place interactions [21]. According to Oliver, loyalty is composed of cognitive loyalty, affective loyalty and conative loyalty [22]. To be specific, tourists' destination loyalty starts from their cognition of the festival attractiveness in the destination; they then establish an affective connection to the destination; finally, the conative loyalty is formed, such as the intention to revisit or a good word-of-mouth referral, etc. However, existing researches have rarely analyzed the relations among festival attractiveness, sense of place and destination loyalty from the cognitive–affective–conative perspective.

To bridge the above-mentioned research gaps, this study aims at using a mixed method approach to survey the festivals in Guangzhou city, China and to achieve the following goals: (1) exploring the dimensions of festival attractiveness through a qualitative method; (2) constituting a conceptual model of "festival attractiveness, sense of place and destination loyalty" based on the cognitive–affective–conative perspective and exploring the influential mechanism of festival attractiveness on destination loyalty through a quantitative method; (3) providing practical basis for the management of festivals and hosting destinations.

This study consists of the following contents. First, relevant literatures on festival attractiveness and destination loyalty are reviewed. Second, a qualitative method is employed to explore the dimensions of festival attractiveness, followed by the conceptualization and measurement of festival attractiveness. Third, the influential mechanism of festival attractiveness on destination loyalty is revealed based on the cognitive–affective–conative perspective. Fourth, this study establishes a connection between festival and destination. This connection not only helps festival organizers find the best point for experiencing destination culture, but it also helps destination marketing organizations find effective means to improve destination loyalty.

## 2. Literature Review

### 2.1. Festival Attractiveness

Attractiveness is a concept that originated from interpersonal psychology. It describes the positive attitude or orientation toward others based on personal expectation, which is subject to the trend of social changes [23]. If marketing is the pushing factor behind tourism, then attractiveness is the pulling factor. Thereupon, tourism attractiveness is the pulling factor formed to attract tourists. According to tourism attractiveness theory, although tourists have strong personal motivations and preferences, their

choices of destinations and activities depend on the diverse resources and properties of the destination as well as their perceptual evaluations on the resources and properties [24]. A festival is a thematic public grand occasion or annual celebration involving tourism, leisure and cultural elements [25], which is an important attractive factor of a destination. Based on the characteristics of attractiveness and festivals, this study defines festival attractiveness as what can both attract tourists to participate in festivals and satisfy their benefit perception and cognitive value.

In existing researches, festival attractiveness is often regarded as one of the indicators for measuring festival attractiveness or destination attractiveness. Among the researches regarding festival attractiveness as festival motivation, Kruger and Saayman studied an arts festival in South Africa and classified festival motivation into escape, innovation, festival property, festival attractiveness and socialization [26]. They also explored the relationship between tourists' behavioral intentions and their motivations through the case of a jazz festival, and classified motivation into festival attractiveness, fun, jazz appreciation, socialization, travel and relaxation [27]. Among the researches that regard festivals as destination attractiveness, Hu and Ritchie pointed out that destination attractiveness includes multiple dimensions, such as scenery, climate and accessibility, as well as festivals and special events [9]. From the perspective of demand, Lee, Ou and Huang explored the factors that influence the attractiveness of Taiwan hot spring tourism destinations, including safety assurance, leisure and entertainment, etc.; among the factors, leisure and entertainment factors include festivals and special events [5].

As discussed above, due to the uniqueness of festivals, festival attractiveness is not completely equal to festival motivation, nor could it be fully replaced by the measuring methods of destination attractiveness. Through the literature review, it is found that researches on festival attractiveness are rare and that there is a lack of theoretical accumulation. Besides, festival attractiveness is determined by both the supplier and the demander rather than by only one of them. Therefore, festival organizers and participators should be included in the further exploration of festival attractiveness.

### 2.2. Sense of Place

Sense of place, as an important concept in the research on people–place affection, is a research topic deeply discussed in human geography. It emphasizes both the characteristics of the place itself and the emotional connections established by the interactions between people and place, which are created through the different meanings people derive from and then attach to places during their environmental experience [21]. Put simply, sense of place is the experience of a place produced through people–place interaction, which is then internalized into human experience. Festivals create, shape and strengthen ideology in a place, and their success lays the foundation for people–place interaction and promotes the formation of sense of place. Lau and Li studied the case of Hong Kong through an in-depth interview, discussing the local residents' understanding of the meaning of festivals and their connection to the construction of local uniqueness. Their study revealed that the three underlying themes of festival meanings, namely religion and heritage, social bonding, and imagined locality, are crucial elements attributing to the sense of place that eventually shape the identification of a unique place [28]. Lau and Li focused on the case of Hong Kong again and employed grounded theory to further investigate the relationship between festival and place. They demonstrated that environmental, social and ideological factors interlace with one another, helping combine the emotional and cognitive meanings of place to form a comprehensive understanding of the relationship between festival and place [29].

Sense of place is a complex and multi-dimensional concept, and its multidimensionality has been studied in many researches on place. Nevertheless, when existing researches tried to constitute the dimensions of sense of place, the dimensional constitutions differed due to different research objects, perspectives, regions and themes. Among the dimensions of sense of place, e.g., meaning of place [30], social connection [31], place dependence, place attachment and place identity [32], place attachment and place identity are the focuses of many researches. Place attachment is the affective connection between a person and a particular place [33], emphasizing the affective part in the relationship between

people and place. Place identity concerns the emotional attachment to physical environments that people regard as part of their self-identity [34]. Place identity plays a more significant role in the subjective construction of place [35].

Jorgensen and Stedman thought that both place attachment and place identity are components in the multi-dimensional construction of sense of place [36]. Davis used an interpretive methodology to explore the process of festival experience; he found that both place attachment and place identity are the main mechanisms constructing the relationship between tourists and tourism environment, and that both of them influence attendees' place-based perceptions [37]. Hence, this study divides sense of place into two dimensions: place attachment and place identity, so as to explore its relationship with festival attractiveness and destination loyalty.

### 2.3. Destination Loyalty

In the marketing field, repeated purchasing and recommendation to others are often referred to as customer loyalty. As Lovelock once said, customer loyalty refers to a situation in which consumers are willing to stay focused on one enterprise, purchase and use its products repeatedly and recommend its products to their friends and colleagues [38]. Later, researchers introduced customer loyalty into tourism products, destination or leisure/entertainment [39,40]. Tourism destination is regarded as a product, and tourists can revisit the destinations or recommend the tourism destinations to other potential tourists (such as friends or relatives) [41]. However, in the tourism industry, tourists have a relatively low-level loyalty to tourism destinations. This is probably due to the comprehensiveness of the tourism industry (i.e., many tourism enterprises need to work together to satisfy tourists' needs) and tourists' desire for new experiences at new destinations [42]. As a result, tourists may not want to revisit the same destination even though the destination had met their expectations.

At present, destination loyalty is measured in various ways and constitutes multiple dimensions. Simpson and Siguaw regarded word-of-mouth referral as the index of destination loyalty [43]. Lee, Kyle and Scott believed that revisit intention, word-of-mouth referral and destination preference are the three most important dimensions constituting destination loyalty [20]. Based on the tourists of heritage tourism in China, Su et al. explored the relations among tourist perception, overall satisfaction and trust, and destination loyalty, during which destination loyalty was measured by positive word-of-mouth referrals and revisit intentions [44]. Thus, this paper regards word-of-mouth referral and revisit intention as the dimensions of tourism destination loyalty.

### 2.4. Cognitive–Affective–Conative Model

Oliver suggested that cognitive loyalty, affective loyalty, conative loyalty and action loyalty are generated in turn [22]. Cognitive loyalty is based on the faith in the superiority of the property of one product over that of the other, which is the weakest form of loyalty [45]. Affective loyalty is based on customers' affective attitude toward a product, which means that customers will generate affective loyalty toward a product once they have a good attitude toward it. The above factors will lead to intentional loyalty, which will eventually turn revisit intentions into actual behaviors. In short, the formation of customer loyalty is a process from perception to affective connection and then to behavioral intention, which can be summarized by the cognitive–affective–conative model [46]. According to this process, in the first stage (cognition generation stage), customers evaluate available information; the second stage (affection development stage) mainly focuses on customers' affection or affective response to experience process; and finally, positive cognition and affective evaluation affect behavioral intention.

Based on the cognitive–affective–conative model, Gracia et al. built a conceptual model of "service quality, affective response and customer loyalty". Through the investigation of hotel and restaurant customers, they found that service quality increases positive affective response and thus improves customer loyalty, and that positive affective response plays a partial mediating role between service quality perceptions and customer loyalty [47]. By utilizing a modified theory of reasoned action

that contains cognitive–affective–conative components, Kim et al. investigated the relations among perceived value, satisfaction and revisit intention [48]. Ahn and Back also explored the brand loyalty in the integrated resort setting from the cognitive–affective–conative perspective. Their conceptual model sets two-way communication, emotional exchange and brand partner quality as the cognitive stage, brand attitude as the affective stage, and behavioral intention as the conative stage. Their research proved that cognitive loyalty is the important antecedent of affective loyalty and that affective loyalty has a significant effect on conative loyalty [49]. Based upon existing researches, we employed the cognitive–affective–conative model to help build a conceptual model of "festival attractiveness, sense of place and destination loyalty", so as to explore the relations among them.

## 3. Research Case and Design

This paper chose Guangzhou city as the case study. Guangzhou is one of the 24 famous cities that were first approved to be national historical and cultural cities by the State Council of China. As a city with both traditional cultures and modern vitality, Guangzhou was also recognized as a world first-tier city by GaWC (Globalization and World Cities). To increase popularity and economic benefits as well as to enrich local residents' lives, many influential festivals have emerged in Guangzhou city. The competitions among these festivals are fierce, resulting in ever increasing scales and internationalized levels. The representatives of traditional festivals include the Spring Festival Flower Fair and the Boluo's Birthday Festival; typical modern festivals are the Guangfu Temple Fair and the International Lightening Festival. Besides the above-mentioned festivals, Guangzhou also holds various festivals such as the International Food Festival and the Qiqiao Festival. These festivals have become important tourist attractions that attract tourists from all over the world. Consequently, this paper suggests that Guangzhou and its festivals are suitable as cases for studying the relationship between festival attractiveness and destination loyalty, for they are typical and accord with the purpose of this study.

To better understand the dimensions of festival attractiveness as well as its relationship with sense of place and destination loyalty, this study used a sequential mixed method approach with a qualitative phase followed by a quantitative phase of data collection and analysis. Based on this, Study One employed a qualitative method during which nine months had been spent on the collection of interview data about the organizers and the tourists at the Spring Festival Flower Fair, the Boluo's Birthday Festival and the Guangfu Temple Fair. The collection dates are shown in Table 1. After data collection, the interview records were analyzed on the basis of grounded theory to extract the initial dimensions of festival attractiveness. Study Two utilized a quantitative method, during which 466 copies of questionnaires had been collected in two months through on-site (the International Lightening Festival and the International Food Festival), online (Sojump) and Wechat channels. The questionnaire data were then analyzed through the use of the structural equation model to test the reliability of the dimensions of festival attractiveness as well as to explore the relations among festival attractiveness, sense of place and destination loyalty. The initial qualitative phase was to explore the dimensions of festival attractiveness that were neglected in previous studies. These dimensions were to be tested later to further help establish its relationship with destination loyalty in the quantitative phase.

**Table 1.** Festival introduction and interview data collection dates.

| No. | Name | Introduction | Collection Date | Data Source |
|-----|------|-------------|-----------------|-------------|
| 1 | The Spring Festival Flower Fair | Traditional festival. The Spring Festival Flower Fair can be traced back to the Song Dynasty. With the full governmental support in recent years, it has been included in the List of Intangible Heritage and became a traditional activity with strong Southern China cultural characteristics. | 2019.2.2–5 | On-site interview data collection |

**Table 1.** *Cont.*

| No. | Name | Introduction | Collection Date | Data Source |
|---|---|---|---|---|
| 2 | Guangfu Temple Fair | Modern festival. The first Guangfu Temple Fair was held in 2011. It is a festival held by the government of Yuexiu District to popularize Guangfu Temple Fair culture. | 2019.2.19–26 | On-site interview data collection |
| 3 | The Boluo's Birthday Festival | Traditional festival. The Boluo's Birthday Festival is a folk temple fair with a history of more than one thousand years, which contains the most typical traditional folk culture elements of Guangzhou. | 2019.3.17–19 | On-site interview data collection |
| 4 | The International Lighting Festival | Modern festival. Guangzhou International Lightening Festival started in 2011. It has become one of the three World Lighting Festivals. | 2019.11.18–27 | On-site interview data collection |
| 5 | Guangzhou Festivals | The International Food Festival, Qiqiao Festival, etc. | 2019.11–2020.2 | Questionnaire collection through online (Sojump) and Wechat channels |

## 4. Study One: Conceptualization and Scale Development of Festival Attractiveness

### 4.1. Method

The main technique for obtaining qualitative information was in-depth interviews with open-ended questions. The lack of qualitative enquiries into festival attractiveness motivated us to use a qualitative phase to reveal this topic. By utilizing both literature analysis and in-depth interview methods, we described and concluded the essence and the dimensions of festival attractiveness from the perspectives of festive characteristics and tourist perception. Also used grounded theory to analyze the text data. Grounded theory is a kind of bottom-up qualitative research approach that concludes and extracts concepts and categories from source material and then develops them into a theory [50]. Since it was put forward, it has received much attention and recognition and been widely used in many fields, such as psychology and tourism. According to the aim of this paper, we used a purposive sampling approach to collect the interview data of the organizers and the tourists at the Spring Festival Flower Fair, the Guangfu Temple Fair and the Boluo's Birthday Festival. We finally determined eight organizers and 31 tourists. The tourists were mainly from Beijing, Heilongjiang, Hunan, Guangdong and Jiangxi, etc., making the samples more diversified and the results more general.

The interview outline was designed upon a literature review and expert panel discussions and mainly included the following questions, such as "What are the main factors that attracted you to participate in this festival?", "Where do you think the attractiveness of this festival may lie?", and "What is your favorite festival activity? And Why?". A flexible semi-structured interview method was employed during the interview, enabling the interviewees to express their understandings about festival attractiveness publicly and honestly. The interviewees were between 19 to 49 years old, and 80% of them had a bachelor's degree or above. We adopted face-to-face interviews, each of which lasted about 20 to 60 min. Finally, according to the research paradigm of Strauss and Corbin Grounded Theory [50], the interview records were analyzed by open coding, axial coding and selective coding. In this coding process, a qualitative research software, NVivo 11, was used to classify, sort and arrange

the interview data. One of the main advantages of this software is that it allows researchers to develop dimensions or sub-dimensions at any time during the analysis process, providing us with the flexibility to reduce, alter or enhance the dimensions of festival attractiveness.

*4.2. Results*

Through open coding and axial coding, we obtained 16 clustering subcategories and four main categories; by selective coding, we determined the constitutive dimensions of festival attractiveness. To test the theoretical saturation, two more tourists were interviewed but no new concept and categories were found. Therefore, the saturation passed the test, indicating that festival attractiveness includes four dimensions: strong festival atmosphere, harmonious interpersonal interaction, distinct cultural symbols and rich festival activities. The coding analysis of the interview textual data using the NVivo 11 program produced frequency results for each category shown in Table 2.

**Table 2.** The constitutive dimensions of festival attractiveness.

| Second Categories | Primary Categories | Frequency | Total | Percentage |
|---|---|---|---|---|
| Strong festival atmosphere | Bustling festival atmosphere | 56 | 126 | 29.86% |
| | Joyous and peaceful festival atmosphere | 27 | | |
| | Unique festival environment | 11 | | |
| | Ritual activities with rich connotation | 32 | | |
| Harmonious interpersonal interaction | Getting people together | 24 | 56 | 13.27% |
| | Sharing the happy life with others | 14 | | |
| | Expecting a beautiful future with others | 9 | | |
| | Enhancing communications with others | 9 | | |
| Distinct cultural symbols | Feeling the good wishes brought by the auspicious symbols | 27 | 136 | 32.23% |
| | Comprehending the connotations of the cultural symbols | 43 | | |
| | Inheriting and developing the cultural symbols | 42 | | |
| | Seeking the symbols in the memory | 24 | | |
| Rich festival activities | Rich contents of activities | 38 | 104 | 24.64% |
| | Various forms of activities | 31 | | |
| | Innovative elements integrated into festival activities | 23 | | |
| | Multi-cultural elements integrated into festival activities | 12 | | |

Strong festival atmosphere refers to the scenes or the atmosphere that can bring people strong feelings through the layouts, decorations, lightings etc. Atmosphere can be classified into visual elements and non-visual elements. Visual elements attract the tourists mainly through the layout/design of the physical environment, while non-visual elements influence tourists' feelings by the creation of a pleasant atmosphere through lighting, music and temperature, etc., so as to ensure their positive experience [51]. Harmonious interpersonal interaction means tourists' face-to-face interactions with others in the specific time and space of the festival. In essence, a festival is a kind of social activity that provides a means for people to interact and share experiences with each other [52], and tourists can get closer and enhance their affective communications during the interactions.

Distinct cultural symbols refer to the carrier that contains the cultural connotation in the festival, through which people express their appeals, wishes, ideals and pursuits. A festival is a platform for

the popularization of destination cultures and a miniature of local cultural characteristics. The rites and interactions, the legends, the mascots, the traditional works of art etc. are all important parts of festival symbols, which embody the tourists' longing for and pursuit of a good life. Finally, rich festival activities are rich and diversified activities carefully planned by the organizers for the tourists. Logically, the richer the activities are, the better the tourists' needs can be satisfied and the stronger the attractiveness [53].

## 5. Study Two: The Relations among Festival Attractiveness, Sense of Place and Destination Attractiveness

### 5.1. Hypotheses Development

Based on the festival attractiveness dimensions found in Study One, Study Two employed a quantitative method to further test the constitutive dimensions of festival attractiveness and explored the relations among festival attractiveness, sense of place and destination loyalty from the cognitive–affective–conative perspective.

### 5.1.1. Impact of Festival Attractiveness on Sense of Place

Although festival attractiveness and sense of place can effectively deal with many management issues in reality, little empirical work has been done on the relationship between them. As previously mentioned, festival attractiveness satisfies tourists' benefit perception and cognitive value. According to Oliver's cognitive–affective–conative model [54], as a kind of cognitive response, perceived festival attractiveness leads to an affective response, which in turn is a predictor of behavior intention [48]. Sense of place describes an affective connection between people and environment, which can help connect cognition to conation [55]. Place attachment and place identity are the important dimensions of sense of place [29,37]. In former researches, destination attractiveness was treated as the antecedent of place attachment [56], and a positive causal relationship exists between them. Festivals are often regarded as a tool to enhance the tourism appeal of a destination, which can also promote the formation of tourists' unique sense of place [29].

However, the formation of sense of place not only relies on the location of a place, but also depends on peoples' participation and experience in the place [57]. As a social symbol of place, festivals offer a place for tourists to experience sense of place, and they also provide an opportunity to inherit and protect the symbolic elements (e.g., histories and customs) and meanings of a place [28]. As Cheng, Wu and Huang pointed out, destinations attract tourists by way of providing high-quality recreational activities or protecting histories and cultures, thus enhancing tourists' attachment to the place [10]. In addition, after studying China's famous historical city Hangzhou, Xu and Zhang found destination attractiveness has a significant effect on tourists' place attachment and that local festivals are important elements constituting destination attractiveness [58].

In festival tourism, organizers provide unique environments with local characteristics to create a strong festival atmosphere for distinct cultural symbols and immersive festival activities, so as to enhance interactions among tourists. Then, tourists will get a positive affective experience when immersed in such an environment. Besides, Lee and Chang studied aboriginal festivals, finding that affective experience promotes the formation of tourists' place identity [59]. All these researches indirectly show that tourists will form attachment and identity to the festival-hosting destination when they are impressed by the characteristics of the festival. Thus, the following hypotheses are proposed:

**Hypothesis 1a,b,c,d (H1a,b,c,d):** *Festival attractiveness has a significantly positive effect on place attachment.*

**Hypothesis 2a,b,c,d (H2a,b,c,d):** *Festival attractiveness has a significantly positive effect on place identity.*

This study suggests that sense of place consists of place attachment and place identity, which are two related but different phenomena. In the study comparing natives and the nonnatives, Hernández,

Carmen Hidalgo, Salazar-Laplace and Hess got the following findings. For the natives, place identity tends to be consistent with place attachment. For the nonnatives, place attachment will be rapidly formed along with time once the interaction with the environment is built, but the formation of place identity is relatively complicated [35]. In brief, place attachment is formed ahead of place identity. Similarly, when investigating the effect of undergraduate students' residential mobility on place attachment and place identity, Vidal et al. also proved that place attachment can promote the formation of place identity [60]. Based on these, the following hypothesis is proposed:

**Hypothesis 3 (H3):** *Place attachment has a significantly positive effect on place identity.*

5.1.2. Impact of Sense of Place on Destination Loyalty

Sense of place is an important concept that explains destination loyalty. Lee et al. employed place attachment to measure the tourists' attitudinal loyalty toward United States National Forests, and they found that attitudinal loyalty (place attachment) and intentional loyalty are highly and positively related [61]. Similarly, Yuksel, Yuksel and Bilim also proved that place attachment is the prerequisite for destination loyalty. They found that the three dimensions of place attachment (place dependence, affective attachment and place identity) have direct effects on cognition and affective loyalty but also have indirect effects on intentional loyalty through the overall satisfaction with the destination [19]. Furthermore, Lee, Kyle and Scott believed that place attachment plays a mediating role between festival satisfaction and destination loyalty. They divided place attachment into two dimensions: social connection/place identity and place dependence; they also divided destination loyalty into three dimensions: revisit intention, word-of-mouth recommendation and destination preference [20]. The results showed that place identity/social connection has a strong positive effect on intention to revisit. Thus, the following hypotheses are proposed:

**Hypothesis 4 (H4):** *Place attachment has a significantly positive effect on word-of-mouth referral.*

**Hypothesis 5 (H5):** *Place identity has a significantly positive effect on word-of-mouth referral.*

**Hypothesis 6 (H6):** *Place attachment has a significantly positive effect on intention to revisit.*

**Hypothesis 7 (H7):** *Place identity has a significantly positive effect on intention to revisit.*

In addition, the proposed model and hypothesized relationships are illustrated in Figure 1. In this model, tourists' festival attractiveness is assumed to be the cognitive stage. In the affective stage, festival attractiveness is regarded as the antecedent of sense of place. Finally, cognitive variable and affective variable are measured to predict tourists' destination loyalty and to analyze the full model that contains all possible relations. Thus, this study tests the above hypotheses.

*5.2. Research Design*

5.2.1. Measurement

The variables designed in this research include festival attractiveness, sense of place and destination loyalty. The contents of the questionnaire are structured, which contains four parts. The first part focuses on tourists' perceived festival attractiveness, which is based on interview data and refers to Kruger and Saayman's scale [27]. The festival attractiveness in the questionnaire includes four dimensions and 16 items. The second part investigates tourists' sense of place, which mainly refers to the scale of Yuksel, Yuksel and Bilim [19]. The third part measures tourists' loyalty toward the festival hosting destination, which refers to the scale of Lee, Kyle and Scott [20]. Finally, the fourth part surveys the demographic features and the basic information of the tourists, including their genders, ages, levels of education and the names of the last festivals they participated in. The variables in the

first three parts are all indicated by a 5-point Likert scale: 1 stands for "Strongly Disagree" and 5 stands for "Strongly Agree".

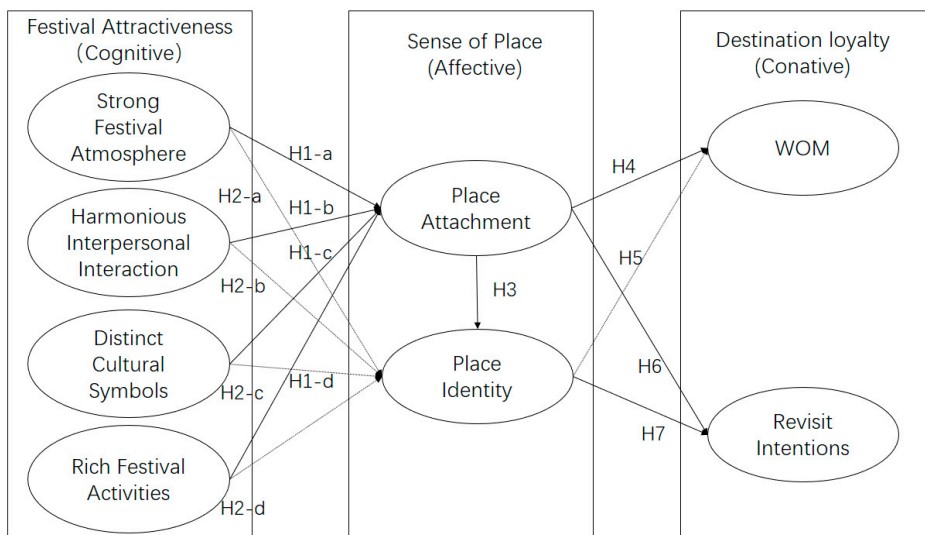

**Figure 1.** The hypothesized relationships among festival attractiveness, sense of place and loyalty.

### 5.2.2. Data Collection and Sample Profile

In order to gather the data of tourists participating in Guangzhou festivals as comprehensively as possible, we collected questionnaires in two stages: an on-site survey and an online survey. The on-site survey was conducted by four trained team members at the International Lighting Festival and the International Food Festival during 18 to 27 November 2019 and 22 November to 1 December 2019, respectively. A total of 260 copies of questionnaires were collected by random sampling. After that, the online survey was conducted through Wechat and Sojump (a third-party survey website) channels. A total of 270 copies of questionnaires were collected. Among the 530 copies of questionnaires collected on-site and online, 466 copies were valid, and thus the validity rate was 87.9%. This is because the questionnaires that were filled quickly, incomplete in information and too invariable in choices were excluded. Among the interviewees, 51.1% were female while 48.9% were male. Most of the interviewees were between 20–29 years old (51.1%), 24.7% were between 30–39 years old, 13.9% were below 19 years of age and 10.3% were above 40 years of age. A majority of them had a bachelor's degree or above (91%). The last festivals that the tourists participated in were the International Lighting Festival (27%), the International Food Festival (23.3%), the Spring Festival Flower Fair (18.5%), the Guangfu Temple Fair (16.7%), the Boluo's Birthday Festival (6.9%), the Qiqiao Festival (6%) and others (1.7%).

### 5.3. Data Analysis and Results

### 5.3.1. Exploratory and Confirmatory Factor Analyses on Festival Attractiveness

The research on festival attractiveness is still in the exploratory stage; thus this study employed SPSS23.0 software to conduct exploratory factor analysis (EFA) on the 16 items of festival attractiveness. As is shown in Table 3, the Kaiser-Meyer-Olkin (KMO) value was found to be 0.795, and the Bartlett's Test of Sphericity resulted in $p < 0.001$. If the eigenvalue is greater than 1, then four factors are found, which respectively correspond to the four dimensions: strong festival atmosphere, harmonious interpersonal interaction, distinct cultural symbols and rich festival activities; the four factors could explain 73.805% of the variation for festival attractiveness, which exceeds the limit of 60%. Thus, the four factors are acceptable.

**Table 3.** Results of reliability and convergent validity analysis of festival attractiveness.

| Dimension | Items of Measurement | Factor Loading | Mean Value | Factor Mean Value |
|---|---|---|---|---|
| Strong Festival Atmosphere (SFA) AVE = 0.654; CR = 0.883 | It has a lively festive atmosphere | 0.835 | 4.23 | 4.27 |
| | It has a joyful and peaceful festival atmosphere | 0.842 | 4.26 | |
| | It has a unique festival environment | 0.855 | 4.26 | |
| | It is rich in ritual activities | 0.847 | 4.34 | |
| Harmonious Interpersonal Interaction (HII) AVE = 0.682; CR = 0.865 | Attending the festival brings us closer together | 0.829 | 3.74 | 3.56 |
| | Attending the festival allows me to share the beautiful life with others | 0.895 | 3.50 | |
| | Attending the festival enhances interaction with others | 0.865 | 3.43 | |
| Distinct Cultural Symbols (DCS) AVE = 0.580; CR = 0.847 | To feel the good wishes of auspicious symbols | 0.822 | 3.91 | 3.68 |
| | To comprehend the connotation of cultural symbols | 0.773 | 3.65 | |
| | To inherit and develop cultural symbols | 0.854 | 3.43 | |
| | To explore the symbols in memory | 0.819 | 3.71 | |
| Rich Festival Activities (RFA) AVE = 0.627; CR = 0.834 | Its activities are rich in content | 0.828 | 3.68 | 3.44 |
| | It has various forms of festival activities | 0.885 | 3.50 | |
| | It combines a variety of local and international cultural elements | 0.851 | 3.13 | |

Note: AVE = average variance extracted, CR = composite reliability.

To further test the rationality of the above structure, this research also conducted confirmatory factor analysis (CFA) by using the AMOS17.0 software to check the validity of the scale. The overall fitting of the CFA model is as follows: $\chi^2/df$ = 1.709, goodness-of-fit index (GFI) = 0.923, comparative fit index (CFI) = 0.967, incremental fit index (IFI) = 0.967, Tucker-Lewis index (TLI) = 0.961, relative fit index (RFI) = 0.910, root mean square residual (RMSEA) = 0.039; every index is on an optimal level. As regards the inner fitting quality, after removing one item each from harmonious interpersonal interaction and rich festival activities, the standardized factor loadings of the remaining 14 items in their respective dimensions are between 0.717 and 0.879; all of them pass the significance test on the level of $p < 0.001$. The composite reliabilities (CR) of all dimensions are greater than 0.7 and the average variances extracted (AVE) are all above 0.5, which indicate good convergent validities [62]. The discriminant validity can be tested by comparing the square roots of the variances extracted with the correlation coefficient of each variable. Since the correlation coefficients of all dimensions are between 0.113 and 0.525, according to the AVE of each dimension, the arithmetic square roots could be obtained, which are between 0.762 and 0.826. All of these arithmetic square roots are greater than the correlation coefficients of the variables, indicating that the scale has an optimal discriminant validity [62].

5.3.2. Structural Model Assessment

In this research, the maximum likelihood estimate (MLE) method in AMOS17.0 software was used to do parameter estimation and to test the degree of model fitting. Then based on all samples, a structural model was built, with festival attractiveness and sense of place being the exogenous variables and destination loyalty being the endogenous variable. According to the data analysis, $\chi^2/df$

= 1.766, GFI = 0.919, CFI = 0.963, IFI = 0.963, TLI = 0.958, RFI = 0.907 and RMSEA = 0.041. These values show that the overall fitting of the structural model is good. As presented in Table 4, the results show that strong festival atmosphere, harmonious interpersonal interaction, distinct cultural symbols and rich festival activities have significantly positive effects on place attachment, which supports H1a, H1b, H1c and H1d; the path coefficients are 0.425 ($p < 0.001$), 0.345 ($p < 0.001$), 0.280 ($p < 0.001$) and 0.164 ($p < 0.01$), respectively. In addition, strong festival atmosphere, harmonious interpersonal interaction and distinct cultural symbols have significantly positive effects on place identity, which supports H2a, H2b and H2c; and the path coefficients are 0.269 ($p < 0.01$), 0.241 ($p < 0.01$) and 0.490 ($p < 0.001$), respectively. Rich festival activities have no significant effect on place identity ($\beta = 0.056$, $p > 0.05$), thus H2-d is invalid. Place attachment has a significantly positive effect on place identity ($\beta = 0.263$, $p < 0.01$), indicating that H3 is valid. Both place attachment ($\beta = 0.360$, $p < 0.001$) and place identity ($\beta = 0.151$, $p < 0.01$) have positive effects on word-of-mouth referral, which supports H4 and H5, respectively. Both place attachment ($\beta = 0.096$, $p < 0.05$) and place identity ($\beta = 0.178$, $p < 0.001$) have positive effects on intention to revisit, which supports H6 and H7, respectively.

**Table 4.** Test results of the model.

| Hypothesis | Path | Standardized Path Coefficients (t-Value) | Results |
|---|---|---|---|
| H1a | Place Attachment <– Strong Festival Atmosphere | 0.425 *** (7.024) | Supportive |
| H1b | Place Attachment <– Harmonious Interpersonal Interaction | 0.345 *** (5.820) | Supportive |
| H1c | Place Attachment <– Distinct Cultural Symbols | 0.280 *** (4.049) | Supportive |
| H1d | Place Attachment <– Rich Festival Activities | 0.164 ** (2.789) | Supportive |
| H2a | Place Identity <– Strong Festival Atmosphere | 0.269 *** (3.261) | Supportive |
| H2b | Place Identity <– Harmonious Interpersonal Interaction | 0.241 ** (3.043) | Supportive |
| H2c | Place Identity <– Distinct Cultural Symbols | 0.490 *** (5.277) | Supportive |
| H2d | Place Identity <– Rich Festival Activities | 0.056 (0.749) | Unsupportive |
| H3 | Place Identity <– Place Attachment | 0.263 ** (3.253) | Supportive |
| H4 | Word-of-Mouth Referral <– Place Attachment | 0.360 *** (5.925) | Supportive |
| H5 | Word-of-Mouth Referral <– Place Identity | 0.151 ** (3.123) | Supportive |
| H6 | Intention to Revisit <– Place Attachment | 0.096 * (2.187) | Supportive |
| H7 | Intention to Revisit <– Place Identity | 0.178 *** (4.867) | Supportive |

Note: * Path coefficient is significant at $p < 0.05$. ** Path coefficient is significant at $p < 0.01$. *** Path coefficient is significant at $p < 0.001$.

## 6. Conclusions and Implications

### 6.1. Conclusions

Many tourism destinations in China have been organizing festivals to attract tourists, turning festivals into important tourist attractions [25]. However, existing researches only treat festival as an important part of destination attractiveness instead of studying festival attractiveness and its relationship with destination loyalty. Therefore, based on the festivals of Guangzhou, this study employed a mixed approach of both qualitative and quantitative methods to conceptualize festival attractiveness and to test its relationships with sense of place and destination loyalty. This study has the following conclusions.

Firstly, based on qualitative research, we determined the four dimensions (strong festival atmosphere, harmonious interpersonal interaction, distinct cultural symbols and rich festival activities) of festival attractiveness and developed a festival attractiveness scale containing 14 items. Although previous researches had pointed out the importance of festival attractiveness to festival motivation and destination attractiveness, this study conceptualized the significant dimensions of festival attractiveness, which adds a new perspective to the existing literature.

Festival attractiveness is different from festival motivation. Festival attractiveness emphasizes the pulling effect of festivals (such as festival atmosphere and festival activities) to the tourists. However, festival motivation is subject both to a pulling effect and a pushing effect [27], for it can be pulled by the characteristics of festivals but also pushed by the intangible driving factors (such as social motivation) among the tourists. Additionally, festival attractiveness cannot be fully replaced by destination attractiveness. Festival attractiveness emphasizes the perception of intangible elements like festival atmosphere, interpersonal interaction, etc., but destination attractiveness focuses more on tangible elements, such as the natural environment, cultural scenic spots, infrastructure, etc. [58]. As was found in this research, the intangible elements in festival atmosphere and interpersonal interaction are important elements that attract the tourists, while these intangible elements need to be brought out through tangible elements in cultural symbols and festival activities. This indicates that the tourists prefer emotional communication in a tangible environment to satisfy their emotional needs by means of festival activities and cultural symbols.

Secondly, after the significant dimensions of festival attractiveness were determined, we developed a festival attractiveness scale and, based on the cognitive–affective–conative model, proposed a model illustrating the relations among festival attractiveness, sense of place and destination loyalty. Through quantitative analysis, we demonstrated that strong festival atmosphere, harmonious interpersonal interaction, distinct cultural symbols and rich festival activities have positive contributions to place attachment. According to this, tourists' perception of the bustling festival and the harmonious atmosphere will produce an affective connection to the hosting destination; their perceptions of distinct cultural symbols and festival activity quality will also promote their understanding of local cultures, thus producing affective connection.

Besides, we also proved that strong festival atmosphere, harmonious interpersonal interaction and distinct cultural symbols are related to place identity. This means that, upon tourists' identification with the meaning of the place, their perceptions of festival atmosphere and cultural symbols (containing local cultures) and interpersonal interaction (stressing social connection) will evolve into place identity. However, rich festival activities have no significant effect on place identity. This is probably because festival activities are rich, diverse, innovative and multi-cultural, and tourists' perception of festival activities is mostly based on the interest and the diversity of the activities instead of their cultural characteristics, so they cannot form place identity. Additionally, tourists' place attachment is related to their place identity, which proves the viewpoint of Hernández, Carmen Hidalgo, Salazar-Laplace and Hess [35] is also valid in the researches of festival. This also proves, on the other hand, that festival attractiveness can promote the formation of place attachment; but not all of the dimensions of festival attractiveness can promote the formation of place identity.

Thirdly, in a festival background, tourists' place attachment and place identity have positive effects on destination loyalty. As regards the two dimensions of destination loyalty, place attachment has a more positive effect on word-of-mouth referral than on intention to revisit. However, place identity has an equal positive effect on intention to revisit and word-of-mouth referral. This means tourists tend to give a positive word-of-mouth referral once they establish an affective connection through the interaction with the festival-hosting destination. If tourists form identification through giving positive meaning to the festival-hosting destination, they will tend to give positive word-of-mouth referrals and be willing to revisit the destination. This idea differs, however, from the findings of Lee, Kyle and Scott [20].

In summary, the theoretical contributions of this study lie in two aspects. (1) Regarding festival attractiveness, the past studies discussed it as one dimension of festival motivation or destination attractiveness [5,9,26,27]. However, we defined festival attractiveness, which had been neglected by existing researches, and determined its four constitutive dimensions through qualitative research; the four dimensions clarified the core difference between festival attractiveness and destination attractiveness. Moreover, we developed and tested the festival attractiveness scale to more accurately describe tourists' perceptions of festival attractiveness. In this sense, our study contributes to the

theoretical development of the festival literature. (2) Previous studies emphasized that festivals may enrich a unique sense of place and contribute to the ontological construction of a place [28,29], while this study empirically examined the relationship between festival attractiveness and sense of place. Based on the cognitive–affective–conative model, this research proposed a relationship model of "festival attractiveness, sense of place and destination loyalty" to establish a close relationship between festival and hosting destination. The study also confirmed the progressive mechanism that festival attractiveness (cognitive loyalty) affects sense of place (affective loyalty), which further affects destination loyalty (intentional loyalty). Our study enriches the theoretical understanding of how festivals impacts tourists' destination loyalty.

### 6.2. Practical Implications

Based on the analysis of various kinds of festival data in Guangzhou, this study summarizes the dimensions of festival attractiveness, which may provide some implications for festivals in other cities. From a practical perspective, the findings of this study can help festival organizers create a festival atmosphere with strong local characteristics, so as to enhance tourists' perceptions of festival attractiveness and to improve their affective connection to the destination as well as their identification with the meaning of the destination. The festival attractiveness scale developed can provide detailed guidance for enhancing tourists' perceptions. To strengthen tourists' interactions and create a bustling but peaceful festival atmosphere, festival organizers can do the following things: (1) further exploring the cultural characteristics and traditional ritual activities of the destinations; (2) exhibiting cultural festival symbols about decorations, rites and diets; (3) organizing diverse festival activities.

In addition, there are an increasing number of destinations like Guangzhou, which attract tourists by hosting a variety of festivals. From the perspective of destination management, the formation procedure of destination loyalty should be recognized. This study offers a new perspective and idea for the management of destination loyalty. The administrators of the destinations should think more about establishing the connection between local festivals and the destinations. Publicity and marketing works should be done to highlight the festival activities of the destinations and their characteristics, so as to attract tourists from all over the world. Meanwhile, the administrators should also provide places for local festivals and improve the infrastructures to give tourists enough space for local cultural experience, thus helping form sense of place and improving destination loyalty.

### 6.3. Limitation and Future Research Directions

As with any research, the current research had several limitations, which may restrict the generalizability of its findings. First, although a festival attractiveness scale was developed and tested, its external validity still needs further determination. This study was limited to the festivals in Guangzhou, so future researches should focus on the data collection of festivals in different types of destinations to further test the scale, so as to improve the external validity of the research findings. Second, this study did not involve tourists from other countries and cultural backgrounds apart from Chinese tourists, so it did not incorporate other cultural structures into the proposed model to investigate the cultural differences. The emphasis of future researches can be laid on the cultural diversity of festival attractiveness and sense of place. Third, although this study empirically tested the relationships among festival attractiveness, sense of place and destination loyalty based on the cognitive–affective–conative model, subsequent studies can incorporate other variables (e.g., festivalscape, festival quality and festival experience) to further confirm the causal relationships between other festival factors and destination. Additionally, action loyalty should also be taken into consideration.

**Author Contributions:** Conceptualization, J.L.; data curation, J.L.; formal analysis, J.L., writing—original draft, J.L.; writing—review and editing, J.L., G.D., J.T. and Y.C. All authors have read and agreed to the published version of the manuscript.

**Funding:** This research was funded by the National Natural Science Foundation of China (grant number 41571132) and Guangdong Key Platforms and Scientific Research Projects (grant number 2017GWQNCX020).

**Conflicts of Interest:** The authors declare no conflict of interest.

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
