# Peer review of "Conceptualizing Festival Attractiveness and Its Impact on Festival Hosting Destination Loyalty: A Mixed Method Approach"

_sustainability, doi:10.3390/su12083082_

Round 1

Reviewer 1 Report

Firstly, I would like to congratulate the authors. I consider that it is an interesting topic and it represents an advance on research. I really enjoy reading the paper.

They offer a model to measure festival attractiveness which represents an advance on literature, and to analyze the relationship with sense of place and destination loyalty.

The paper uses a mixed technique to collect information, which is the best option to create good measurement scales.

But there is need some recommendations for improvement.

  • In the case of qualitative research, I do not understand well the number of participants, because on the epigraph 4.1 Method, the authors mention 8 organizers and 31 tourists, however in table 2, some frequencies are higher that 39.
  • In addition, the authors mention that they use grounded theory, however they do not mention the software used to analyze the information and extract the dimensions.
  • The authors do no include the items they use to measure place attachment, place identity, WOM and revisit intentions.
  • I would like to know why the authors have not checked the relationship between festival attractiveness and destination loyalty, as festival attractiveness satisfies tourists´ benefit perception.

Reviewer 2 Report

Dear colleagues,
thank you very much for the opportunity to review your article about conceptualizing festival attractiveness and its impact on festival hosting destination loyalty. I appreciate its professional processing and the usage of several methods, whose combination and correct application supported the credibility of the results of your research. I consider as a weakness of your article the local applicability of the research results.  Congratulations to you and I wish you much success in your further research activity.

Best regards

Reviewer 3 Report

It should be better to highlight the state of art and the gap between the previous research and your study's findings.

Reviewer 4 Report

Dear authors,

The manuscript analyses festivals' attractiveness from a different perspective, which means that the subject and the contents of the manuscript are relevant and up to date. However, in my opinion, festival's attractiveness is one dimension of destination attractiveness but I agree that this is an acceptable point of view and research.

Globally, the manuscript is well structured and addresses several and important aspects concerning the attractiveness in the tourism context. The methodology planned seems adequate, however, the methodology application appears to be confusing.

My recommendation for you, is to do a better explanation of why the usage of the instruments (interview and questionnaire), due to the fact that it is not clear. Rewrite in a simple way Study one (4) and Study two (5), because the way they are quite confusing and consequently difficult to read and to understand.

Concerning table 1 and 2 add horizontal lines or line break between the different rows, because it is hard to understand where the different rows start

The references used are appropriate.

Round 2

Reviewer 4 Report

The authors made some of the suggested revisions and the article improved globally.